# PAK1 as a Potential Therapeutic Target in Male Smokers with EGFR-Mutant Non-Small Cell Lung Cancer

**DOI:** 10.3390/molecules25235588

**Published:** 2020-11-27

**Authors:** Jae Heun Chung, Taehwa Kim, Yong Jung Kang, Seong Hoon Yoon, Yun Seong Kim, Sung Kwang Lee, Joo Hyung Son, Bongsoo Son, Do Hyung Kim

**Affiliations:** 1Department of Internal Medicine, Pusan National University Yangsan Hospital, Yangsan 50612, Korea; jhchung7942@gmail.com (J.H.C.); taehwagongju@naver.com (T.K.); david7942@hotmail.com (Y.J.K.); drysh79@gmail.com (S.H.Y.); yskim@pusan.ac.kr (Y.S.K.); 2Department of Thoracic and Cardiovascular Surgery, Pusan National University Yangsan Hospital, Yangsan 50612, Korea; drlsk@naver.com (S.K.L.); pnumed09@naver.com (J.H.S.); wtknight98@gmail.com (B.S.)

**Keywords:** non-small cell lung cancer, p21-activated kinase 1, prognosis, male, smokers, EGFR

## Abstract

P21-activated kinases (PAKs) are serine/threonine protein kinases that contribute to several cellular processes. Here, we aimed to determine the prognostic value of PAK1 and its correlation with the clinicopathological characteristics and five-year survival rates in patients with non-small cell lung cancer (NSCLC). We evaluated PAK1 mRNA and protein expression in NSCLC cells and resected tumor specimens, as well as in healthy human bronchial epithelial cells and adjacent healthy lung tissues, respectively, for effective comparison. Immunohistochemical tissue microarray analysis of 201 NSCLC specimens showed the correlation of PAK1 expression with clinicopathological characteristics. The mRNA and protein expression of PAK1 were 2.9- and 4.3-fold higher in six of seven NSCLC cell types and human tumors (both, *p* < 0.001) than in healthy human bronchial epithelial BEAS-2B cells and adjacent healthy lung tissues, respectively. Decreased survival was significantly associated with PAK1 overexpression in the entire cohort (χ^2^ = 8.48, *p* = 0.0036), men (χ^2^ = 17.1, *p* < 0.0001), and current and former smokers (χ^2^ = 19.2, *p* < 0.0001). Notably, epidermal growth factor receptor (EGFR) mutation-positive lung cancer patients with high PAK1 expression showed higher mortality rates than those with low PAK1 expression (91.3% vs. 62.5%, *p* = 0.02). Therefore, PAK1 overexpression could serve as a molecular target for the treatment of EGFR mutation-positive lung cancer, especially among male patients and current/former smokers.

## 1. Introduction

Lung cancer is the leading cause of cancer-related deaths [1]. Estimates show that 221,347 new diagnoses of cancer and 82,344 deaths occurred in Korea in 2019 [2]. Furthermore, non-small cell lung cancer (NSCLC), classified as adenocarcinoma, squamous cell carcinoma, and large cell carcinoma, accounts for nearly 80% of all lung cancers [3].

Recent progress in the development of therapies targeting epidermal growth factor receptor (EGFR), anaplastic lymphoma kinase rearrangements, and rare oncogenic drivers (such as ROS1, RET, MET, and EGFR exon 20 insertion mutations) has considerably improved the survival in a subset of patients with NSCLC [4,5]. Additionally, novel molecular targets for NSCLC treatment are required to improve lung-cancer-related mortality rates.

P21-activated kinases (PAKs) contribute to cellular motility [6], differentiation [7], proliferation [8], and survival [9]. The downstream effector PAK1 belongs to the Rho family of small GTPases that include Cdc42 and Rac1, which are essential regulators of cellular migration and invasion and promote cancer metastasis [10,11,12].

PAK1 inhibition was observed to induce cell cycle arrest via the accumulation of lung cancer cells in the G1 phase [13], and significantly impair tumor growth in lung cancer xenograft models, indicating that PAK1 drives NSCLC tumor proliferation [14,15]. However, further studies are warranted to determine whether PAK1 expression in NSCLC tumors correlates with the recurrence-free and overall survival of patients.

We postulated that PAK1 plays an important role in lung carcinogenesis and compared PAK1 expression in cells from seven NSCLC and normal epithelial cell lines, including BAES-2B cells, 28 frozen human NSCLC specimens, and paired healthy lung tissue specimens. We also assessed PAK1 expression in association with clinicopathological parameters, including the EGFR mutation status, recurrence-free survival, and five-year survival of 201 patients with banked NSCLC tissue specimens using tissue microarray (TMA) analysis.

## 2. Results

### 2.1. PAK1 Expression in NSCLC Cells/Fresh-Frozen NSCLC Specimens Was Higher than That in BEAS-2B Cells/Adjacent Healthy Lung Tissue Specimens

The prognostic significance of PAK1 was determined by Western blotting and reverse transcriptase polymerase chain reaction (RT-PCR) in vitro and in human tissue specimens. Western blotting revealed higher PAK1 expression in most NSCLC cells than in BEAS-2B cells (Figure 1A). The level of PAK1 protein expressed was 4.3-fold higher in frozen NSCLC specimens than in adjacent healthy tissue specimens (*p* < 0.001; Figure 1B and Appendix A). PAK1 mRNA expression in the 28 frozen specimens measured using RT-PCR was 2.9-fold higher for the NSCLC specimens than for the adjacent healthy tissue specimens (*p* < 0.001; Figure 1B). These results were further supported by public meta- and comparative analyses using Lung Cancer Explorer. Meta-analysis data from six previous studies and TCGA lung cancer data revealed elevated PAK1 expression in NSCLC cells compared to healthy tissues (odds ratio (OR): 1.66 and 1.90 for lung adenocarcinoma (LUAD) and lung squamous cell carcinoma (LUSC), respectively; Appendix A). We also compared PAK1 mRNA overexpression in NSCLC cells and healthy tissues that had similar profiles in the meta-analysis (Appendix A). The results indicate that PAK1 was overexpressed in the NSCLC cells and human tissues.

### 2.2. Association of PAK1 Expression with Histological NSCLC Subtypes and Clinicopathological Characteristics of Patients

We performed immunohistochemical staining of 201 NSCLC resected tumor specimens for the TMA analysis to determine whether PAK1 expression was correlated with the clinicopathological characteristics of patients with NSCLC. The tumor specimens were classified as positive or negative based on a PAK1 staining intensity of ≥0.5 or 0, respectively. PAK1 expression was positive and negative in 111 (55.2%) and 90 (44.8%) of the 201 TMA specimens, respectively. Figure 2A depicts representative PAK1 staining in the TMA specimens. The results of the Wilcoxon rank-sum test showed significantly higher mean PAK1 immunostaining scores for LUSC specimens than for LUAD and bronchioloalveolar carcinoma specimens (1.03 vs. 0.57; Figure 2B). Table 1 summarizes the correlation between PAK1 expression and the clinical characteristics of patients with NSCLC.

### 2.3. Clinical Characteristics of Patients with EGFR Mutations Based on PAK1 Expression

Mutations in the gene encoding EGFR are critical because this receptor is used as a target in lung cancer treatment. Notably, EGFR mutations are highly prevalent among non-smoking Asian females. The results observed in the present study are similar to those reported previously. We observed that sex, smoking, LUAD, and adenocarcinoma tissue (T) classification significantly differed between EGFR-negative and -positive patients (female vs. male: 84.2% vs. 46.8%, *p* < 0.001; smoking: 74.3% vs. 31.0%, *p* < 0.001; LUAD: 46.7% vs. 97.9%, *p* < 0.001; T classification: *p* = 0.003). Additionally, the five-year survival rates significantly differed between patients with EGFR-negative and -positive tumors (39.04 ± 21.6 vs. 51.59 ± 15.2 months, *p* < 0.001; Table 1).

Table 2 shows the baseline characteristics depending on the PAK1 status of the patients harboring EGFR mutations. Among patients with EGFR-positive tumors, 24 and 23 tested positive and negative for PAK1 expression, respectively. The prevalence of female sex and smoking was lower in the PAK1-positive than in PAK1-negative patient populations (female vs. male: 62.5% vs. 30.4% *p* = 0.028; smoking: 45.8% vs. 21.7%, *p* = 0.081). The two groups did not differ significantly with respect to age, histological type, TNM classification, differentiation, or lymphovascular invasion. However, the five-year mortality rates were lower in PAK1-negative than in PAK1-positive patients (62.5% vs. 91.3%, *p* = 0.020). We verified this result in vitro as well. The PAK1 and phospho-PAK1 levels were elevated in HCC827 (EGFR-mutated adenocarcinoma cell line) cells compared to those in A549 (EGFR wild-type cell line) or BEAS-2B (normal bronchial epithelial cell line) cells (Appendix A). We further observed that gefitinib-resistant HCC827 (HCC827/GR) cells had higher levels of phosphorylated PAK1 than wild-type HCC827 cells (Appendix A). We used NVS-PAK1-1, a PAK1 selective inhibitor, to suppress the PAK1 and phospho-PAK1 levels in HCC827/GR cells, and found that both PAK1 and phospho-PAK1 levels were suppressed in a dose-dependent manner (Appendix A). In the assessment of cell viability, the inhibition of PAK1 using NVS-PAK1-1 led to cell death (Appendix A). These results indicate that PAK1 may be associated, at least partially, with the survival rate in patients with EGFR-mutant lung cancer.

### 2.4. Effects of PAK1 Overexpression on Five-Year and Recurrence-Free Survival

The Kaplan–Meier survival curves (Figure 3A) indicate that PAK1 overexpression was associated with a lower probability of five-year survival. PAK1 overexpression (immunostaining score ≥0.5) was significantly associated with decreased five-year survival among patients with NSCLC (χ^2^ = 8.48, *p* = 0.0036; Figure 3A). Univariate Cox regression analysis showed that sex (OR, 0.328; *p* = 0.001), smoking status (OR, 2.846; *p* = 0.001), histological subtype (OR, 2.506; *p* = 0.002), and pathological TNM stage (OR, 2.009; *p* = 0.017) were factors that significantly affected the overall survival of patients (Table 3). The Kaplan–Meier survival curves showed that PAK1 overexpression was not associated with recurrence-free survival in 159 patients diagnosed with early-stage (≤IIIA) NSCLC (χ^2^ = 0.158, *p* = 0.695; Figure 3B). We further evaluated whether PAK1 overexpression would be associated with a greater number of patients with lung cancer using a Kaplan–Meier plotter, which contained information from 1925 patients with lung cancer derived from data available in the PAK1 Affy database (ID: 202161_at; Appendix A). Our findings showed that PAK1 mRNA overexpression was associated with poor overall survival in patients with lung cancer (hazard ratio (HR), 1.28; *p* = 0.00011, log-rank test).

### 2.5. Effects of PAK1 Expression on Five-Year Survival Based on Sex and Smoking Status

Based on the findings from the TMA analysis, the inverse relationship between PAK1 expression and survival significantly depended on the sex and smoking status of the 201 patients. PAK1 overexpression was associated with a significantly lower probability of survival among males (χ^2^ = 17.1, *p* < 0.0001; Figure 3C). Similar findings were noted among smokers with PAK1 immunostaining scores that were inversely correlated with five-year survival. Additionally, PAK1 overexpression was significantly associated with an increased risk of death (χ^2^ = 19.2, *p* < 0.0001; Figure 3D).

## 3. Discussion

Consistent with the results of recent studies, which showed that PAK1 expression was upregulated in lung cancer compared to that in adjacent healthy bronchial tissues [11,12], we found that NSCLC specimens had higher PAK1 levels than healthy bronchial cells. However, there were certain discrepancies in terms of PAK1 protein expression and mRNA levels in this study. Recently, several microRNAs involved in the modulation of PAK1 molecules were identified. Studies also indicate that the microRNAs regulate cell growth and invasion by directly targeting PAK1 in multiple solid tumors [16,17]

The Kaplan–Meier survival plots showed that in the 201 NSCLC specimens included in the TMA analysis, PAK1 overexpression was significantly associated with a decreased five-year survival of patients. The effects of PAK1 overexpression on survival depended on sex and smoking status. The effect of PAK1 expression on the five-year survival of current or former smokers (defined as patients who had smoked at least 100 cigarettes in their lifetime, but had quit smoking by the time of the hospital visit) resulted in elevated chi-square scores compared to those obtained for patients who were non-smokers and showed PAK1 expression (χ^2^ = 8.48 vs. 19.2). The results of our immunohistochemical studies on NSCLC TMAs revealed significantly higher mean PAK1 immunostaining scores in squamous cell carcinoma than in adenocarcinoma tissue specimens. Since smoking is a well-known risk factor in squamous cell carcinomas, but not in adenocarcinomas, the relatively high PAK1 expression in squamous cell carcinoma in this study could be related to the possibility of enhanced PAK1 expression owing to smoking. With respect to the smoking status, the trend among male patients with NSCLC was similar, as their chi-square scores increased by approximately two-fold. These data suggest that PAK1 expression is a useful predictor of five-year survival among male patients with NSCLC who are smokers.

The reason why PAK1 overexpression is a negative prognostic factor for patients who have been exposed to smoke remains unclear. Perhaps, the functional polymorphisms in the PAK1 gene (rs2154754) are responsible for the significant influence of smoking levels on lung cancer risk [18]. Moreover, PAK1 regulates the expression of FRA-1, a proto-oncogene whose expression is induced by cigarette smoke [19]. These data suggest that certain confounding factors affecting former and current smokers can help overcome the effects of PAK1 expression, and consequently affect the five-year survival of patients.

PAK1 overexpression was not found to be significantly associated with sex-based differences among patients with gastric [20] or bladder [21] cancer. However, we observed that PAK1 expression depended on sex and played an important role in the five-year survival of patients with NSCLC. Our study on lung cancer cell lines and bioinformatic analysis showed that PAK1 expression differed based on sex and exhibited a correlation with lung fibrosis associated with the protease-activated receptor 2 pathway (data not shown).

We observed that PAK1 overexpression was not associated with recurrence-free survival in patients with early-stage NSCLC. This finding contradicts those from other studies that assessed the same in solid breast cancer, colorectal carcinoma tumors, and renal cell carcinoma [22,23,24]. However, in the studies on breast and colorectal cancers, the *p*-value of the association between PAK1 levels and recurrence-free survival rate was 0.05, which is statistically ambiguous. In addition, since earlier studies on patients with breast cancer reported the role of the combination of PAK1 and CCND1 expressions as a predictor of recurrence-free survival, the relationship between PAK1 alone and the recurrence-free survival rate is unclear.

Increased PAK1 expression did not correlate significantly with parameters such as the N and M classification of the TNM stage or lymphovascular invasion, which are related to cell invasion and migration. P21-activated kinase 1 regulates the expression of the actin cytoskeleton during cell motility and invasion, and it might be associated with cancer development [25,26,27]. The contradictory results obtained with respect to the role of PAK1 in tumor migration determined from real-world data might be attributable to experimental limitations. However, this could be addressed using tumor specimens obtained from surgically resectable tumors and adjacent healthy bronchiolar tissues. Therefore, further studies should focus on the influence of PAK1 on invasion in advanced or metastatic NSCLC.

Recent developments in targeted therapies have improved the survival rates of patients with lung cancer. However, patients treated with EGFR tyrosine kinase inhibitors usually acquire resistance after 9–12 months. Previous studies on EGFR mutations showed that a PAK1 inhibitor, administered in combination with a selective inhibitor of protein kinase C, was effective against the acquisition of resistance induced by EGFR tyrosine kinase inhibitors [28,29]. We found that the five-year mortality rates were lower in patients with PAK1-negative than in PAK1-positive EGFR mutant lung cancer. Therefore, further studies are warranted to better understand the role of the EGFR/PAK1 pathway for improving the survival rates in patients with EGFR-mutated NSCLC, even after they have acquired resistance to tyrosine kinase inhibitors.

In summary, the present study provided evidence that supported the role of PAK1 in NSCLC pathogenesis. First, PAK1 was found to be overexpressed in NSCLC cells and in banked NSCLC specimens. Second, PAK1 overexpression was independently correlated with significantly decreased overall survival in patients with NSCLC, although the extent depended on the smoking status and sex. Third, patients with high PAK1 expression and EGFR mutant lung cancer showed poor prognosis and survival. Therefore, targeting PAK1 using PAK1 inhibitors might be a potential therapeutic strategy for treating EGFR-mutant lung cancer, especially among male smokers.

## 4. Materials and Methods

### 4.1. NSCLC Tissue Specimens and TMA Construction

We obtained 29 frozen, surgically resected tumor and adjacent healthy lung tissue specimens, derived from patients who were treated by lobectomy for primary NSCLC, from the tissue bank at Pusan National University Yangsan Hospital. The 201 NSCLC specimens were histologically examined and classified using the 2015 World Health Organization International Classification System for Lung Tumors as adenocarcinomas or bronchioloalveolar carcinomas (*n* = 120) and squamous cell carcinomas (*n* = 81) [30]. Thereafter, tumor TMAs were constructed based on two tissue cores (each with a 2 mm diameter) derived from the central and peripheral areas of the tumors. The institutional review board of the Pusan National University Yangsan Hospital approved the use of the archived clinical tissue specimens (05-2018-162), and written informed consent was obtained from all patients for the use of specimens.

### 4.2. Antibodies and Compounds

We purchased antibodies and reagents from the following suppliers: polyclonal rabbit anti-human PAK1 antibody (cat. no: 2602; Cell Signaling Technology, Danvers, MA, USA), anti-rabbit IgG–horseradish peroxidase-conjugated antibody and monoclonal antibody against β-actin (Santa Cruz Biotechnology Inc., Dallas, TX, USA), selective PAK1 inhibitor NVS-PAK1-1 (cat. no: HY-100519), gefitinib (cat. no: HY-50895; MedChemExpress LLC., Monmouth Junction, NJ, USA), PRO-PREP™ Protein Extraction Solution (iNtRON Biotechnology, Seongnam-Si, Korea), TRIzol reagent (Invitrogen, Carlsbad, CA, USA), and amfiRivert cDNA Synthesis Platinum Master Mix (GenDEPOT, Katy, TX, USA). All other reagents were of the purest grade available.

### 4.3. Cells and Culture Conditions

Healthy BEAS-2B human epithelial lung and H1703 human squamous carcinoma cells (American Type Culture Collection, Manassas, VA, USA), and A549, HCC827, HCC2108, HCC95, HCC1588, and HCC1195 NSCLC cells (Korean Cell Line Bank, Seoul, Korea) were cultured as follows. BEAS-2B cells were cultured using the Bronchial Epithelial Cell Growth Medium Bullet Kit (Lonza Group Ltd., Basel, Switzerland). NSCLC cells were maintained in RPMI-1640 (WELGENE Inc., Gyeongsan-si, Korea) supplemented with 10% heat-inactivated fetal bovine serum, glutamine (2 mM), penicillin (100 U/mL), and streptomycin (100 µg/mL).

### 4.4. Western Blotting

The human lung tissue samples were homogenized using a TissueLyser II (Qiagen, Hilden, Germany) at 30 Hz for 1 min and lysed in PRO-PREP™. Total protein concentrations were measured using Bio-Rad protein assays (Bio-Rad Laboratories Inc., Hercules, CA, USA), as instructed by the manufacturer. Western blotting was performed for the protein analysis as described in [31], and the concentrations were measured by densitometry using Image J (National Institutes of Health). The results were normalized to the levels of β-actin.

### 4.5. RT-PCR

Total RNA (2 μg) extracted from human lung cells and tissues using TRIzol reagent was reverse-transcribed using the amfiRivert cDNA Synthesis Kit, as instructed by the manufacturer. The sense and antisense primers used were as follows: PAK1, 5′-CGTGGCTACATCTCCCATTT-3′ and 5′-TCCCTCATGACCAGGATCTC-3′; GAPDH, 5′-CAGCCTCAAGATCATCAGCA-3′ and 5′-TGTGGTCATGAGTCCTTCCA-3′, respectively. The transcribed DNA was amplified by PCR using the Bioneer RT/PCR PreMix under the following conditions: 32 cycles of 20 s at 95 °C, 20 s at 60 °C, and 30 s at 72 °C. The amplicons were separated by electrophoresis in 2% agarose gels and visualized using Dyne LoadingSTAR. Signal intensity was quantified using Image J.

### 4.6. Immunohistochemistry

NSCLC, adjacent healthy tissue, and TMA NSCLC tissue specimens were stained immunohistochemically using the anti-PAK1 antibody (1:200) according to manufacturer’s instructions and visualized using the cellSens standard DP21 (Olympus Life Sciences, Tokyo, Japan). After nuclear immunostaining, quantification was performed using light microscopy with an Eclipse Ts2 microscope (Nikon, Tokyo, Japan). The staining intensity of each sample was assigned a score ranging from 0 to 3+, with 0, 1+, 2+, and 3+ indicating absent, weak, moderate, and strong staining, respectively. No interobserver discrepancies were detected in the scores. In general, the stained tissue sections were studied and scored separately by two pathologists who were blinded to the clinical parameters. A third pathologist arbitrated in the event of disagreements.

### 4.7. Analysis of Datasets

Correlations between PAK1 mRNA expression and overall survival were determined using the Kaplan–Meier plotter (http://kmplot.com/analysis). A database was established using gene expression and survival data (http://kmplot.com/analysis/index.php?p=service&cancer=lung) derived from 1926 patients with NSCLC. Briefly, PAK1 expression data were entered into the database to obtain Kaplan–Meier survival plots in which the number-at-risk was indicated below the main plot. Between-group differences were evaluated using HR with 95% confidence intervals and log-rank *p*-values. The lung cancer-specific database Lung Cancer Explorer contains information collected from over 6700 patients and 56 studies (http://lce.biohpc.swmed.edu/lungcancer/index.php#page-top). Gene expression in patients with lung cancer can be evaluated by meta- and comparative analyses using this database. We compared the levels of PAK1 expression in LUAD and LUSC tumors to those in normal controls.

### 4.8. Statistical Analysis

Baseline characteristics are expressed in terms of mean ± SD. The significance of differences between groups was analyzed using Student’s *t*-test and one-way ANOVA with Dunnett’s test. Between-group differences in survival (in months) were evaluated using Kaplan–Meier plots, and the significance of differences was determined using log-rank tests. Recurrence-free survival was defined as time elapsed from date of surgery to clinically defined recurrence, including local, regional, and distant recurrence. Significant risk factors and ORs were determined using univariate Cox regression. The effects of PAK1 expression on EGFR mutations were evaluated using subgroup analyses. Differences with *p* values <0.05 were considered statistically significant.

## Figures and Tables

**Figure 1 molecules-25-05588-f001:**
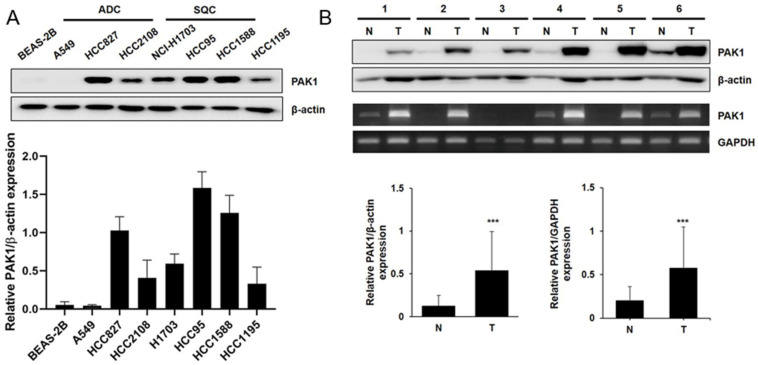
P21-activated kinase 1 (PAK1) expression in vitro and in human lung tumors. (**A**) Total cell lysates from healthy human lung epithelial BEAS-2B cell cultures; cells from three adenocarcinoma (ADC), three squamous cell carcinoma (SQC), and adeno-squamous cell carcinoma HCC1195 cell lines were subjected to Western blotting for measuring PAK1 expression. β-actin was used as an internal control. The results were expressed as means ± SD values from three separate experiments. Significance was determined using one-way ANOVA (*p* < 0.001 vs. BEAS-2B cells). (**B**) Proteins obtained from human adenocarcinoma tissue (T) and pair-matched healthy tissue (N) specimens were analyzed using Western blotting. The total mRNA extracted from adenocarcinoma and paired healthy tissue specimens was examined by reverse transcriptase polymerase chain reaction (RT-PCR) analysis. GAPDH served as an internal control. Values are expressed as means ± SD (*n* = 28 replicates). Significance was calculated using the Student’s *t*-test (*** *p* < 0.001 vs. healthy tissue group).

**Figure 2 molecules-25-05588-f002:**
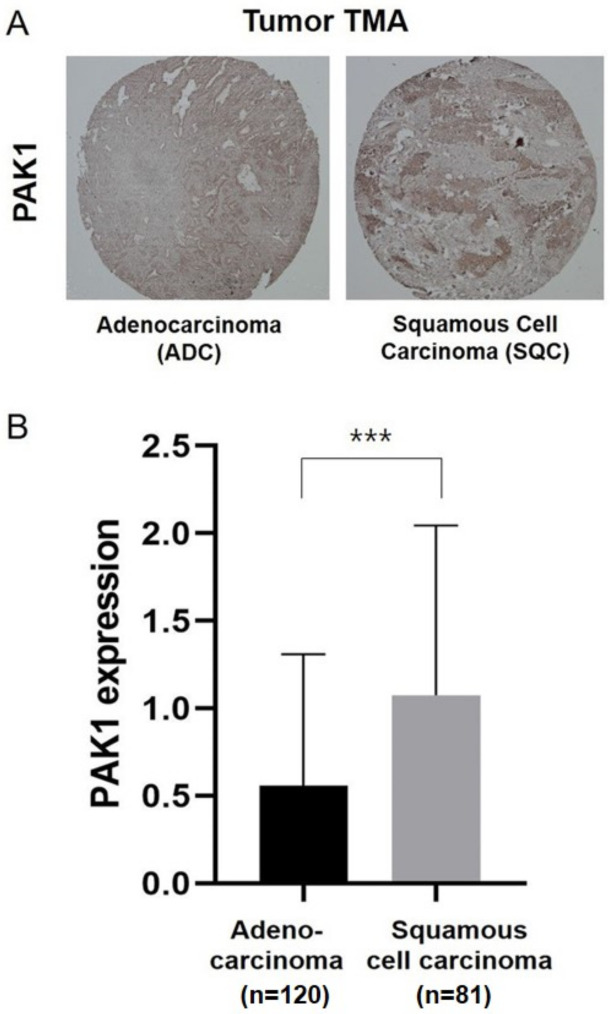
Immunohistochemical analysis of PAK1 expression in lung tumor tissue. (**A**) ADC and SQC images recorded at a magnification of 20×; (**B**) distribution of the staining scores for the two histological non-small cell lung cancer (NSCLC) subtypes. Note: PAK1 immunostaining was performed using tissue microarray (TMA) specimens obtained from 201 patients with NSCLC (120 with adenocarcinoma and 81 with squamous cell carcinoma). *** *p* < 0.001

**Figure 3 molecules-25-05588-f003:**
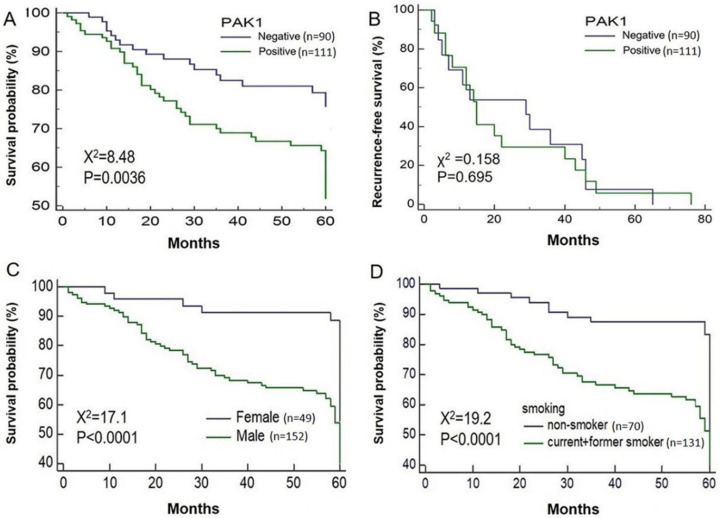
Kaplan–Meier curves showing five-year and recurrence-free survival according to PAK1 expression, based on cohort, sex, and smoking status. (**A**) The mean five-year survival duration of patients based on PAK1 expression. Five-year survival duration for PAK1-positive patients was shorter than that of PAK1-negative patients (46.2 months vs. 52.5 months, respectively, *p* = 0.036). (**B**) Recurrence-free survival of patients of early-stage NSCLC based on PAK1 expression. (**C**) Five-year survival curve based on PAK1 expression in male patients. (**D**) Five-year survival curve based on PAK1 expression in smokers. Note: Specimens from 201 patients were studied to determine the five-year survival based on the presence or absence of PAK1 expression.

**Table 1 molecules-25-05588-t001:** Characteristics of patients with NSCLC categorized according to PAK1 and epidermal growth factor receptor (EGFR) expression.

	PAK1 Negative (%) (N = 90)	PAK1 Positive (%) (N = 111)	*p* Value	EGFR Mutation Negative (%) (N = 154)	EGFR Mutation Positive (%) (N = 47)	*p* Value
Age (median)	65.6 ± 10.2	64.1 ± 10.8	0.305	66.1 ± 10.1	60.5 ± 11.0	0.001
Sex (male)	58 (64.4)	94 (84.7)	0.001	130 (84.4)	22 (46.8)	<0.001
Smoking (current + former)	47 (52.2)	84 (75.7)	0.001	115 (74.7)	16 (34.0)	<0.001
Histologic subtype (adenoca vs squamous)	64 (71.1) vs. 26 (28.9)	56 (50.5) vs. 55 (49.5)	0.003	74 (48.1) vs. 80 (51.9)	46 (97.9) vs. 1 (2.1)	<0.001
Pathologic TNM stage (I vs. II+III+IV)	59 (65.6) vs. 31 (34.4)	54 (48.6) vs. 57 (88)	0.016	79 (51.3) vs. 75 (48.7)	34 (72.3) vs. 13 (27.7)	0.011
T classification			0.051			0.003
T1	50 (55.6)	45 (40.5)		64 (41.6)	31 (66.0)	
T2	35 (38.9)	49 (44.1)		71 (46.1)	13 (27.7)	
T3	5 (5.6)	14 (12.6)		18 (11.7)	1 (2.1)	
T4	0 (0)	3 (2.7)		1 (0.6)	2 (4.3)	
N classification			0.488			0.170
N0	64 (71.1)	68 (61.3)		95 (61.7)	37 (78.7)	
N1	15 (16.7)	23 (20.7)		33 (21.4)	5 (10.6)	
N2	10 (11.1)	19 (17.1)		24 (15.6)	5 (10.6)	
N3	1 (1.1)	1 (0.9)		2 (1.3)	0 (0)	
M classification (M0 vs. M1)	87 (96.7) vs. 3 (3.3)	100 (90.0) vs. 11 (9.1)	0.100	144 (94.1) vs. 9 (5.9)	43 (91.5) vs. 4 (8.5)	0.523
5-year survival (month)	43.8 ± 20.0	40.9 ± 21.7	0.319	39.3 ± 21.6	51.6 ± 15.2	<0.001
Differentiation (well to moderate vs. poor)	71 (78.9) vs. 19 (21.1)	80 (72.1) vs. 31 (27.9)	0.266	110 (71.4) vs. 44 (28.6)	41 (87.2) vs. 6 (12.8)	0.028
Lymphovascular invasion	67 (74.4)	79 (71.2)	0.605	47 (30.5)	8 (17.0)	0.069

**Table 2 molecules-25-05588-t002:** Subgroup analysis based on PAK1 expression in patients with EGFR-mutated NSCLC.

	PAK1 Negative (%) (N = 24)	PAK1 Positive (%) (N = 23)	*p* Value
Age (median)	58.4 ± 12.1	62.8 ± 9.3	0.172
Sex (male)	15 (62.5)	7 (30.4)	0.028
Smoking (current + former)	11 (45.8)	5 (21.7)	0.081
Histologic subtype (adenoca vs. squamous)	23 (95.8)	23 (100)	0.322
T classification			0.378
T1	15 (62.5)	16 (69.6)	
T2	6 (25.0)	7 (30.4)	
T3	1 (4.2)	0 (0)	
T4	2 (8.3)	0 (0)	
N classification			0.816
N0	18 (75.0)	19 (82.6)	
N1	3 (12.5)	2 (8.7)	
N2	3 (12.5)	2 (8.7)	
N3	0 (0)	0 (0)	
M classification			0.780
(M0 vs. M1)	21 (87.5) vs. 3 (12.5)	22 (95.7) vs. 1 (4.3)	
Differentiation			0.232
Well to moderate	21 (37.5)	20 (65.2)	
Poor	3 (12.5)	3 (12.5)	
Lymphovascular invasion	4 (16.7)	4 (17.4)	0.947
5-year mortality rate	15 (62.5)	21 (91.3)	0.020

**Table 3 molecules-25-05588-t003:** Cox regression analysis for associated prognostic factors in five-year survival.

Variables	5-Year Survival			
	No. of Patients	Odd Ratio	95% Confidential Interval	*p* Value
Sex (male)	152/201	0.328	0.167 ± 0.643	0.001
Smoking (current + former)	131/201	2.846	1.563 ± 5.180	0.001
Histologic subtype (adenoca)	119/201	2.506	1.392 ± 4.514	0.002
Pathologic TNM stage (I vs. II + III + IV)	113/201	2.009	1.133 ± 3.561	0.017
T classification (T1 vs. T2 + T3 + T4)	94/201	1.753	0.999 ± 3.074	0.050
M classification (M0 vs. M1)	187/201	3.190	0.862 ± 11.805	0.082

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
