# Peer review of "PAK1 as a Potential Therapeutic Target in Male Smokers with EGFR-Mutant Non-Small Cell Lung Cancer"

_molecules, 2020, doi:10.3390/molecules25235588_

Round 1
Reviewer 1 Report
The current study tries to explore the PAK1 as a potential therapeutic target in EGFR-mutant non-small cell lung cancer patients, especially in male smokers. In this study authors have shown that indeed PAK1 levels are elevated both in vitro (cells) and in vivo (patients) compared to their normal counterparts. Authors also reported a correlation between PAK1 expression with the clinicopathological parameters including EGFR-mutation status, recurrence-free survival. In addition, elevated levels of PAK1 independently correlated with significantly decreased overall survival in patients with NSCLC, although the extent depended on their smoking status and sex. Altogether authors suggest that treating EGFR-mutant NSCLC using PAK1 inhibitors might represent a potential therapeutic strategy, especially among male smokers.
The study design, experimental approaches, results and discussion in the study is clear and concise to an extent and requires clarifications in some areas in order improve the quality of manuscript for potential publication
Major points:
While authors succinctly shown that PAK1 levels are elevated in tumor cells and patients’ tissues, I observed that it’s not the rule indeed exceptions exist in the samples (Figure 1B).
For example, samples number 10-12, 20 and 23, PAK1 levels were not elevated. Also, in few patients’ samples, the protein expression levels does not correlate with mRNA levels (for examples sample number 3, 8, 10-12 etc., (Figure 1B and Supplementary Figure 1)
It would be great if authors can discuss these discrepancies in terms of PAK1 expression among patients’ samples, and proteins vs mRNA levels
In the section 2.2,
authors discussed that they collected 201 NSCLC specimens and analyzed for PAK1 expression levels. Among 201, 111 (55.2%) positive and 90 (44.8%) negative. But again, in Figure 2B they presented that 119 samples are adenocarcinoma, and 82 are squamous cell carcinoma. I am confused here? What are the positive or negative for PAK1 levels then?
Why again 119 adenocarcinoma samples are low for PAK1 levels compared to 82 (squamous cell carcinoma). How authors define these parameters?
In the materials and methods, it mentioned that 116 are adenocarcinoma and 85 are squamous cell carcinoma. It would be good in the readers point of view, if Authors should be clear on this number and classification.
In the section 2.3 (Line number 112-114),
Additionally, the 5-year mortality rates significantly differed between patients with EGFR-negative and -positive tumors (39.04 ± 21.6 vs. 51.59 ± 15.2 months, P < 0.001; Table 1). But in the table, it was annotated as 5yr survival rate. Authors should be clear on this.
The objective of a meta-analysis study is to develop a single conclusion that has a greater statistical power. However, authors stated in the paper that (Line number 209-210, 214-215, 220-223) “This conclusion warrants confirmation by more case-control and cohort studies”. The authors’ conclusions clearly indicate that more studies are required to develop correlation between PAK1 expression in male vs female, smoker’s vs nonsmokers, recurrence free survival in EGFR positive NSCLC. The alternative use of "weasel words" throughout the manuscript might help. Regardless of how it is done, the gap between a strict interpretation of the data and the stated conclusions needs to be closed.
In the discussion, authors mentioned that PAK1 regulates FRA-1 expression. It would be great strength to this study, if authors show expression levels of FRA-1 (protein or mRNA) in cell lines and tumor patient’s samples.
Minor points,
Supplementary Figure 2,
Whatever analysis done Supplementary Figure 2A & 2B is related to mRNA levels?
In few sections I see different font style, line number 231-232, please try to maintain same font style throughout
Author Response
Major points:
Point 1: While authors succinctly shown that PAK1 levels are elevated in tumor cells and patients’ tissues, I observed that it is not the rule indeed exceptions exist in the samples (Figure 1B). For example, samples number 10-12, 20 and 23, PAK1 levels were not elevated. Also, in few patients’ samples, the protein expression levels does not correlate with mRNA levels (for examples sample number 3, 8, 10-12 etc., (Figure 1B and Supplementary Figure 1) It would be great if authors can discuss these discrepancies in terms of PAK1 expression among patients’ samples, and proteins vs mRNA levels
Response 1: Thank you for pointing this out. In subsequent studies on the molecular mechanism adopted by PAK1, we observed that miRNAs played an important role in PAK1 protein expression. For example, in case of a tumor sample from a patient (23), high mRNA expression did not correlate with protein expression. We found that the levels of miRNA-7letc were increased in the tumor sample #23 compared to those in adjacent normal tissue. In addition, although we used the tumor and adjacent tissue stored in an LN2 gas tank with proper guidelines suggested by the biobank at the institution, there may have been molecular degradation. Our biobank guidelines were updated to improve the quality of samples. We have added this as a potential mechanism in the Discussion as suggested (lines 183-189)
Point 2: In the section 2.2, Authors discussed that they collected 201 NSCLC specimens and analyzed for PAK1 expression levels. Among 201, 111 (55.2%) positive and 90 (44.8%) negative. But again, in Figure 2B they presented that 119 samples are adenocarcinoma, and 82 are squamous cell carcinoma. I am confused here. What are the positive or negative for PAK1 levels then?
Response 2: This was an error on our part. We observed that there was a mistake in Figure 2b. The number of samples of each histologic type had to be modified. A sample was considered PAK1-positive if the staining intensity was greater than 0, and negative otherwise. In this figure, we summed all staining scores obtained from the tissues regardless of positive or negative staining and found that the staining intensity of PAK1 was higher in squamous cell lung carcinoma specimens. We have corrected the number of patients with each histologic type in Figure 2b (total number of adenocarcinoma specimens 120 vs squamous cell carcinoma specimens 81) and line 100.
Point 3: Why again 120 adenocarcinoma samples are low for PAK1 levels compared to 81(squamous cell carcinoma). How authors define these parameters?
Response 3: In a previous study (Cell Physiol Biochem, 2018, 50,304–316), we found that the staining intensity and nuclear transfer of PAK1 were higher in squamous cell carcinoma specimens than in adenocarcinoma specimens. In this study, an increase in PAK1 staining intensity was shown to be associated with smoking. Since smoking is a well-known risk factor for squamous cell carcinomas than for adenocarcinomas, the relatively high PAK1 expression in squamous cell carcinoma specimens observed in this study could be traced to the possibility of increased PAK1 expression resulting from smoking habit. We have addressed this in the Discussion (lines 196-202). We hope you find this explanation suitable.
Point 4: In the materials and methods, it mentioned that 116 are adenocarcinoma and 85 are squamous cell carcinoma. It would be good in the readers point of view, if Authors should be clear on this number and classification.
Response 4: Thank you for pointing out this error. We have corrected the number of specimens of each histologic type in the Materials and Methods (lines 257-258)
Point 5: In the section 2.3 (Line number 112-114), Additionally, the 5-year mortality rates significantly differed between patients with EGFR-negative and -positive tumors (39.04 ± 21.6 vs. 51.59 ± 15.2 months, P < 0.001; Table 1). But in the table, it was annotated as 5yr survival rate. Authors should be clear on this.
Response 5: Thank you for pointing out the error. “5-year survival rate” is the appropriate term in this study. We have corrected the error in section 2.3 (line 113)
Point 6: The objective of a meta-analysis study is to develop a single conclusion that has a greater statistical power. However, authors stated in the paper that (Line number 209-210, 214-215, 220-223) “This conclusion warrants confirmation by more case-control and cohort studies”. The authors’ conclusions clearly indicate that more studies are required to develop correlation between PAK1 expression in male vs female, smoker’s vs nonsmokers, recurrence free survival in EGFR positive NSCLC. The alternative use of "weasel words" throughout the manuscript might help. Regardless of how it is done, the gap between a strict interpretation of the data and the stated conclusions needs to be closed.
Response 6: Thank you for the insightful comments for improving our manuscript. As mentioned, we agree that words that impart ambiguity should be omitted from the manuscript (line 216-217). In addition, we have explained in greater detail the statistical insignificance of PAK1 in recurrence-free survival (line 218-225)
Point 7: In the discussion, authors mentioned that PAK1 regulates FRA-1 expression. It would be great strength to this study, if authors show expression levels of FRA-1 (protein or mRNA) in cell lines and tumor patient’s samples.
Response 7: Thank you for this suggestion. We agree with your comment. We aim to design a study for explaining why carcinogenesis in squamous cell lung carcinoma is more susceptible to smoking than that in adenocarcinoma by demonstrating PAK1/FRA-1 signaling. However, we could not perform the same in this study since we have focused on the clinical aspects of PAK1 to highlight the possible association between PAK1 expression and smoking habit. We hope that our intent is clear from this.
Minor points:
Point 8:In few sections I see different font style, line number 231-232, please try to maintain same font style throughout.
Response 8: Thank you for the comment; we have corrected this discrepancy.
Reviewer 2 Report
The manuscript by Jae Heun Chung et al. correlates the overexpression of PAK1 in NSCLS cell lines and resected tumor cell specimens, thus demonstrating that PAK1 is implicated in the poor prognosis of smokers in particular EGFR-mutattion positive lung cancer patients.
The manuscript is well structured, but requires extensive editing of English language. The most important problem with this paper is that it does not add anything new to the published literature.
In fact, the results obtained by the authors are already well known to the scientific community. Here are some examples:
https://www.nature.com/articles/srep34933
https://www.ncbi.nlm.nih.gov/pmc/articles/PMC5479255/?report=reader
https://biosignaling.biomedcentral.com/articles/10.1186/s12964-019-0446-z
The authors of the above manuscripts have intensively described the role of PAK1 in NSCLS, by correlating PAK1 expression with lung cancers, EGFR expression and smoking patients.
My revision would have been very different if the authors had presented data regarding (for example) new compounds or drug combinations capable of altering the expression / activity of PAK1 in their cellular and patient models. In this case the manuscript would have become also much more inherent to the journal.
Unfortunately, at present, I consider the manuscript not worthy of being published on molecules
Author Response
Point 1: The manuscript is well structured but requires extensive editing of English language.
Response 1: The manuscript was edited by Editage (https://www.editage.com/). Post-revision, the manuscript was re-sent to Editage for further polishing of language.
Point 2: The most important problem with this paper is that it does not add anything new to the published literature. The previous studies have intensively described the role of PAK1 in NSCLC, by correlating PAK1 expression with lung cancers, EGFR expression and smoking patients.
Response 2: We understand your take on the novelty of this study. We believe the following points will help clarify our intent.
(1) We showed the possible role of PAK1 expression in male patients with squamous cell carcinoma who were smokers. Higher levels of PAK1 expression were observed in such patients. As smoking habit is a well-known risk factor for patients with squamous cell carcinomas that for those with adenocarcinomas, our data suggest that PAK1 could serve as an important target molecule associated with smoking-induced squamous cell lung carcinomas. We have addressed this in the Discussion (Line 196-202)
(2) Despite the negative results obtained, our study showed that PAK overexpression was not associated with recurrence-free survival in lung cancer, which contradicts the findings from other studies on breast cancer, colorectal carcinoma, and renal cell carcinoma.
(3) We have provided additional data that validates the association between PAK1 expression and EGFR mutation by indicating that treatment with NVS-PAK1-1, a PAK1 inhibitor, in EGFR TKI-resistant cells decreased viability (provided in the supplementary section). (line 124-133)
We hope these changes have made the manuscript suitable for consideration.
Round 2
Reviewer 2 Report
The authors have implemented the manuscript according to the observations indicated by the reviewer. The manuscript, while not presenting major novelty on the subject, is still worthy of being published in its present form